# Exploring the Multifaceted Nexus of Uric Acid and Health: A Review of Recent Studies on Diverse Diseases

**DOI:** 10.3390/biom13101519

**Published:** 2023-10-13

**Authors:** Masanari Kuwabara, Tomoko Fukuuchi, Yuhei Aoki, Einosuke Mizuta, Motoshi Ouchi, Masafumi Kurajoh, Tatsuya Maruhashi, Atsushi Tanaka, Nagisa Morikawa, Kensuke Nishimiya, Naoyuki Akashi, Yoshihiro Tanaka, Naoyuki Otani, Mihoko Morita, Hiroshi Miyata, Tappei Takada, Hiroshi Tsutani, Kazuhide Ogino, Kimiyoshi Ichida, Ichiro Hisatome, Kohtaro Abe

**Affiliations:** 1Department of Cardiology, Toranomon Hospital, 2-2-2-Toranomon, Minato, Tokyo 105-8470, Japan; 2Laboratory of Biomedical and Analytical Sciences, Faculty of Pharma-Science, Teikyo University, Itabashi, Tokyo 173-8605, Japan; fukuuchi@pharm.teikyo-u.ac.jp; 3Department of Cardiorenal and Cerebrovascular Medicine, Faculty of Medicine, Kagawa University, Takamatsu 761-0793, Kagawa, Japan; aokiyuhei127@gmail.com; 4Department of Cardiology, Sanin Rosai Hospital, Yonago 683-8605, Tottori, Japan; einytori@gmail.com; 5Department of Health Promotion in Nursing and Midwifery, Innovative Nursing for Life Course, Graduate School of Nursing, Chiba University, Chiba 260-8672, Chiba, Japan; ouchi@dokkyomed.ac.jp; 6Department of Pharmacology and Toxicology, School of Medicine, Dokkyo Medical University, Mibu 321-0293, Tochigi, Japan; 7Department of Metabolism, Endocrinology and Molecular Medicine, Graduate School of Medicine, Osaka Metropolitan University, Osaka 5454-8585, Osaka, Japan; masafumi-kurajoh@omu.ac.jp; 8Department of Regenerative Medicine, Research Institute for Radiation Biology and Medicine, Hiroshima University, Hiroshima 734-8553, Hiroshima, Japan; 55maruchin@gmail.com; 9Department of Cardiovascular Medicine, Saga University, Saga 849-8501, Saga, Japan; tanakaa2@cc.saga-u.ac.jp; 10Division of Cardio-Vascular Medicine, Department of Internal Medicine, Kurume University School of Medicine, Kurume 830-0011, Fukuoka, Japan; nagisa1981417@gmail.com; 11Department of Community Medicine, Kurume University School of Medicine, Kurume 830-0011, Fukuoka, Japan; 12Department of Cardiovascular Medicine, Tohoku University Hospital, Sendai 980-8574, Miyagi, Japan; kensuke118@cardio.med.tohoku.ac.jp; 13Division of Cardiovascular Medicine, Jichi Medical University Saitama Medical Center, Saitama 330-8503, Saitama, Japan; noexcuse87@hotmail.com; 14Division of Epidemiology, Graduate School of Public Health, Shizuoka Graduate University of Public Health, Shizuoka 420-0881, Shizuoka, Japan; yoshi.tanaka0626@gmail.com; 15Cardiovascular Center, Dokkyo Medical University Nikko Medical Center, Nikko 321-1298, Tochigi, Japan; naoyuki@dokkyomed.ac.jp; 16Department of Hematology and Oncology, University of Fukui Hospital, Eiheiji 910-1193, Fukui, Japan; mtakai@u-fukui.ac.jp; 17Department of Pharmacy, The University of Tokyo Hospital, Bunkyo, Tokyo 113-8655, Japan; hmiyata924@gmail.com (H.M.); tappei-tky@umin.ac.jp (T.T.); 18National Hospital Organization Awara Hospital, Awara 910-4272, Fukui, Japan; tsutani.hiroshi.tq@mail.hosp.go.jp; 19Department of Cardiology, Japanese Red Cross Tottori Hospital, Tottori 680-8517, Tottori, Japan; ko-36@umin.ac.jp; 20Tokyo University of Pharmacy and Life Sciences, Hachioji, Tokyo 192-0392, Japan; ichida@toyaku.ac.jp; 21National Hospital Organization Yonago Medical Center, Yonago 683-0006, Tottori, Japan; hisatome@med.tottori-u.ac.jp; 22Department of Cardiovascular Medicine, Graduate School of Medical Sciences, Kyushu University, Fukuoka 812-8582, Fukuoka, Japan; abe.kotaro.232@m.kyushu-u.ac.jp

**Keywords:** uric acid, lifestyle, cardiometabolic diseases, neurological diseases, transporters

## Abstract

The prevalence of patients with hyperuricemia or gout is increasing worldwide. Hyperuricemia and gout are primarily attributed to genetic factors, along with lifestyle factors like consuming a purine-rich diet, alcohol and/or fructose intake, and physical activity. While numerous studies have reported various comorbidities linked to hyperuricemia or gout, the range of these associations is extensive. This review article focuses on the relationship between uric acid and thirteen specific domains: transporters, genetic factors, diet, lifestyle, gout, diabetes mellitus, metabolic syndrome, atherosclerosis, hypertension, kidney diseases, cardiovascular diseases, neurological diseases, and malignancies. The present article provides a comprehensive review of recent developments in these areas, compiled by experts from the Young Committee of the Japanese Society of Gout and Uric and Nucleic Acids. The consolidated summary serves to enhance the global comprehension of uric acid-related matters.

## 1. Introduction

The worldwide prevalence of patients with hyperuricemia or gout is increasing [1,2,3]. Genetic factors are recognized as contributors to hyperuricemia and gout, along with lifestyle factors such as consuming a purine-rich diet, alcohol and/or fructose intake, and physical activity. Hyperuricemia is a well-established causative risk factor for gout flares. Additionally, gout flare is associated with a higher risk of cerebrocardiovascular diseases [4,5]. Therefore, lifestyle modifications are recommended for every individual with hyperuricemia or gout.

The activity of uricase, an enzyme catalyzing the conversion of uric acid to allantoin, was lost about 8–20 million years ago, and therefore, uric acid is the end product of purine metabolism in humans. [6] In addition, the renal tubules reabsorb most uric acid filtered in the glomeruli, resulting in 5 to 10 times higher concentrations of serum uric acid in humans than those in other mammalians. These findings suggest that uric acid was necessary for human evolution. Experimental studies have shown that uric acid is a powerful antioxidant. [7] Uric acid may exert its beneficial effects by protecting cells from oxidative damage by maintaining superoxide dismutase, scavenging radical species, and chelating transition metals. [8,9] However, the worldwide prevalence of patients with hyperuricemia or gout is increasing.

While medical treatment for patients with a history of gout and hyperuricemia is recommended, the landscape for treating asymptomatic hyperuricemia remains intricate and variable, with recommendations differing across countries due to insufficient evidence [10,11,12,13]. Although a standardized international consensus for treating asymptomatic hyperuricemia has not been established yet, a wealth of studies has revealed various comorbidities intertwined with hyperuricemia or gout. This review article focuses on the intricate relationship between uric acid and thirteen specific domains: transporters, genetic factors, diet, lifestyle, gout, diabetes mellitus, metabolic syndrome, atherosclerosis, hypertension, kidney diseases, cardiovascular diseases (CVD), neurological diseases, and malignancies. An overview of this review is shown in Figure 1.

The blue lines indicate a high degree of established relationship, while the light blue lines suggest that the relationship has not yet been clearly established.

A comprehensive review of recent developments in each of these domains has been meticulously compiled by the professional members of the Young Committee of the Japanese Society of Gout and Uric and Nucleic Acids. This summary of evidence will prove instrumental in shaping forthcoming discussions and fostering a deeper understanding of this intricate landscape.

## 2. Transporters

Because of its lipophobicity, uric acid cannot passively permeate the lipid bilayer. Therefore, it is reasonable to assume that membrane transporters play a pivotal role in regulating serum uric acid levels. Since identifying uric acid transporter-1 (URAT1) as the first transporter involved in uric acid reuptake from urine in 2002 [14], researchers have made great efforts to find transporters regulating uric acid homeostasis. Owing to these efforts, glucose transporter 9 (GLUT9) and ATP-binding cassette transporter G2 (ABCG2) were discovered as physiologically important uric acid reuptake and efflux transporters, respectively. However, these membrane proteins are not enough to explain the whole picture of uric acid regulation in our body; thus, scientists continue their studies to clarify the unknown factors. Considering that URAT1 is the molecular target of uricosuric agents such as benzbromarone and dotinurad, the discovery of novel transporters could lead to the development of a therapeutic strategy for abnormal uric acid homeostasis. This section focuses on recent advances in the attempts to identify novel uric acid transporters (Figure 2).

### 2.1. Glucose Transporter 12 (GLUT12)

GLUT12 is a member of the glucose transporter family and has been shown to have glucose transport activity in vitro [15]. The relationship between GLUT12 and uric acid was first identified in a genome-wide association study (GWAS) that focused on serum uric acid levels [16]. Based on this GWAS report, Toyoda et al. examined the uric acid transport activity of GLUT12 using in vitro transient overexpression cell systems and confirmed that GLUT12 functions as a uric acid transporter [17]. Furthermore, this study investigated the physiological importance of Glut12 in uric acid homeostasis using a knockout (KO) mouse model. Since mice can metabolize uric acid to allantoin by uricase (Uox), a uric acid metabolizing enzyme, double KO (DKO) mice of *Glut12* and *Uox* genes (*Glut12-Uox* DKO mice) were established and analyzed. As a result, plasma uric acid levels of *Glut12-Uox* DKO mice were significantly higher than those of *Uox* single KO mice. On the other hand, liver uric acid levels and the ratio of liver to plasma uric acid levels were lower in *Glut12-Uox* DKO mice, suggesting the role of Glut12 in transporting uric acid from plasma to the liver. Given that GLUT12 was identified in a GWAS of serum uric acid levels, it appears that GLUT12 would also be a physiologically important uric acid transporter in humans.

### 2.2. Organic Anion Transporter 10 (OAT10)

OATs are involved in the transmembrane transport of organic anions and belong to a wide range of transporter families, including organic cation transporters and organic cation/carnitine transporters, alongside the well-known uric acid transporter URAT1. Although the uric acid transport activities of OAT10 were reported in in vitro studies [18], its physiological importance in uric acid homeostasis had not been reported for a decade. This was mainly because mutations in the *OAT10* gene were rare in individuals other than Japanese, preventing the detection of this gene in GWAS. Recently, to clarify the roles of OAT10 in uric acid homeostasis, comprehensive exon sequencing analyses were conducted in Japanese gout patients and healthy controls [19]. This study revealed that the allele frequency of missense mutation in the *OAT10* gene, 1129C > T, which induces the amino acid substitution Arg377Cys (R377C), was lower in gout patients. In vitro functional analyses demonstrated that the OAT10 R377C mutant had no uric acid transport activity. A subsequent study revealed that the fractional excretion of uric acid to urine (FE_UA_) was significantly high in people with the OAT10 R377C mutation [20]. Combining the genetic and in vitro analyses, OAT10 was identified as a uric acid reabsorption transporter from urine. Interestingly, some uricosuric agents inhibited the uric acid transport activity of OAT10 in vitro, suggesting the potential of OAT10 as a molecular target of hyperuricemia [20].

### 2.3. Sodium-Dependent Vitamin C Transporter 1/2 (SVCT1/SVCT2)

SVCTs are human homologs of the nucleobase–ascorbate transporter (NAT) family. It is well known that SVCT1/SVCT2 transport vitamin C (VC) in a sodium-dependent manner. Analyses of *Svct1* KO mice revealed the involvement of Svct1 in the reuptake of VC from urine and the regulation of plasma VC levels [21]. In contrast, Svct2 is considered to regulate VC levels in various tissues, such as the liver and kidney, although *Svct2* KO mice die soon after birth [22]. Based on a report demonstrating that the bacterial NAT family transporter YgfU transported uric acid, investigations into the uric acid transport activities of SVCT1/SVCT2 were conducted using in vitro transient overexpression cell systems [23,24]. As a result, it was clarified that both SVCT1 and SVCT2 transport not only VC but also uric acid in a sodium-dependent manner. In mouse models, Svct1 was suggested to regulate serum uric acid levels as a uric acid reabsorption transporter from urine [23]. However, due to lethality, there are no reports investigating changes in uric acid homeostasis in *Svct2* KO mice. Future studies are needed to reveal the physiological importance of SVCT1/SVCT2 in uric acid homeostasis in humans.

## 3. Genetic Factors

Mendelian randomization analyses have provided evidence that genes responsible for hyperuricemia are not independently associated with hypertension, ischemic heart disease, type 2 diabetes, cerebrovascular disease, or heart failure [25,26,27]. These findings suggest that the genes responsible for uric acid regulation, specifically those encoding uric acid transporters, may not directly act as independent risk factors for these health issues. However, it is worth emphasizing that not all genetic factors related to hyperuricemia exert the same influence. Some studies have indicated a positive association between the xanthine oxidoreductase (XOR) gene and blood pressure, hinting at the possible involvement of XOR and oxidative stress in influencing blood pressure levels [28,29]. Additionally, evidence shows a connection between serum uric acid levels and a genetic risk score based on eight uric acid-regulating single nucleotide polymorphisms associated with cardiovascular death and sudden cardiac death [30]. Moreover, Mendelian randomization research using data from UK Biobank and clinical trials has suggested that higher serum uric acid levels may indeed contribute to increased blood pressure, potentially mediating an increased risk of CVD [31].

Uric acid levels are regulated by both uric acid transporters and XOR, which play roles in its accumulation and production. Importantly, while most studies of the uric acid transporter gene were negative, most studies exploring gene regulation of XOR have been positive. In addition, it is crucial to recognize that hyperuricemia is influenced not only by genetic predisposition but also by lifestyle factors, including diet, alcohol and fructose intake, and physical activity. Dietary habits have rapidly changed in the last 100 years, and these acquired factors possibly affect CVD [32]. Therefore, when assessing the risk of CVD related to uric acid levels, we must consider both genetic and acquired factors, taking a comprehensive approach to understand their interplay.

## 4. Diet

Hyperuricemia is a lifestyle-related disease; hence, advising patients to modify their lifestyle is important, regardless of pharmacotherapy. A healthy diet tailored to each individual according to lifestyle and coexisting diseases must be chosen rather than relying solely on a low-purine diet alone. Long-term adherence is important for improving overall health, managing metabolic comorbidities, and preventing and managing conditions such as hyperuricemia or gout.

Healthy diets, such as dietary approaches to stop hypertension (DASH) and Mediterranean diets, have been reported to reduce serum uric acid levels and gout incidences [33]. Studies suggest that a healthy diet combined with weight loss in overweight or obese individuals significantly reduces cardiometabolic risk factors, including blood pressure, cholesterol profile, triglycerides, and insulin resistance, and improves gout outcomes. Consumption of certain dietary risk factors (e.g., alcohol, sugar-sweetened beverages, and red meat) and adherence to a healthy diet are associated with serum uric acid levels and the prevalence of gout; however, it has been shown that these have minimal effects compared to genetic variation [34,35]. Only recently was the interaction between DASH diet adherence and gout risk in women found to have a significant additive gene–diet interaction [36]. Shirai et al. reported that habitual coffee consumption reduced gout risk without altering serum uric acid levels [37] like anti-inflammatory therapy. Similarly, a healthy diet may also have anti-inflammatory effects.

Conversely, the impact of short-term dietary factors such as consuming purine-rich foods and alcohol on gout flare-ups must be heeded. However, the idea of restricting protein consumption to reduce purine load is not accurate. Kaneko et al. reported on purines in foodstuffs [38]; based on the reported values obtained by calculating the optimal energy intake (1800 kcal) and macronutrient energy distribution for 28 days in a hospital diet, the purine quantity is approximately 190–600 mg/day, with a confirmed average value of approximately 380 mg/day. However, it is slightly high at 600 mg/day for a menu including fish such as cutlass [39]. Significant deviation from the recommended quantities is unlikely during energy intake optimization, even if high-purine foods are incorporated throughout a daily diet.

Purine nucleotides are also umami components. Using nucleotide umami substances, such as guanosine 5’-monophosphate and inosine 5’-monophosphate, in combination with monosodium glutamate, has been reported to reduce salt intake without impairing taste [40]. Furthermore, the intestinal epithelium has a high demand for nucleotides for energy acquisition, proliferation, and innate immunity, and the significant increase in nucleotide substrate requirement during injury, infection, and wound healing is well known. It has been recently reported that gut microbiota can utilize multiple purines, including uric acid, as carbon and energy sources and can act as the major source of purines used for nucleotide production by the intestinal mucosa [41]. Various lactic acid strains and other bacteria have been proven to lower serum uric acid levels and improve gout flare-ups in human trials [42]. Gut microbiota changes associated with dietary changes may improve host uric acid metabolism; hence, future research on these aspects is awaited.

## 5. Lifestyle (Children and Adults)

Hyperuricemia is associated with obesity and lifestyle diseases in adults. Gout, the most common presentation of hyperuricemia in adults, is rare in children, and most children with gout have some underlying disease such as Down syndrome, renal hypoplasia, atrial septal defect, glycogen storage disease, leukemia, and methylmalonic acidemia [43]. Therefore, abnormal serum uric acid concentration in children is considered a biochemical disorder with no clinical significance. However, recent evidence has suggested that hyperuricemia in children is an important lifestyle-related clinical problem [44,45,46,47,48]. Some recent topics of uric acid research related to lifestyle are introduced below.

Hyperuricemia in children is associated with obesity, metabolic syndrome, and its components of metabolic syndrome [43]. Because serum uric acid concentrations change with growth in children, with differences between males and females appearing around 10 years of age [44], it is important to establish age- and sex-specific pediatric reference values when defining hyperuricemia in children. In a recent large-scale population-based study of Japanese children aged 9–10, hyperuricemia was found to be associated with obesity, high hemoglobin A1c (HbA1c) levels, dyslipidemia (hypertriglyceridemia and hypo high-density lipoprotein (HDL)-cholesterolemia), and liver damage [44]. In this study, factors associated with hyperuricemia in children were more accurately assessed by focusing on a limited age group. Non-alcoholic fatty liver disease (NAFLD) [45] and non-alcoholic steatohepatitis [46], the main causes of liver injury in children, are associated with elevated uric acid levels. Longitudinal studies have shown that hyperuricemia is a risk factor for the future development of hypertension [47] and chronic kidney disease (CKD) [48].

Because younger patients have fewer complications, studies in those patients with hyperuricemia are reasonable. Further, longitudinal studies are needed to determine the long-term prognosis of patients with hyperuricemia and the effectiveness of interventions.

In adults, lifestyle habits that prevent hyperuricemia include exercise, smoking cessation, and work participation. According to the 2021EULAR guidelines [49], people with rheumatic and musculoskeletal diseases that include hyperuricemia should avoid physical inactivity; they should engage in regular exercise according to their abilities. People with hyperuricemia should be encouraged to stop smoking and be informed that smoking is detrimental to symptoms, function, disease activity, disease progression, and the occurrence of comorbidities. Work participation may have beneficial effects on health outcomes of people with hyperuricemia. However, the relationship between exercise habits and hyperuricemia disappeared when the body mass index (BMI) was adjusted in the model, indicating that the exercise effect was entirely mediated through BMI [35,50]. Conversely, in two population-based cross-sectional studies [51,52], levels of physical activity and sedentary behavior were significantly associated with hyperuricemia status, even with adjustment for BMI. The important thing to remember is that lifestyle improvements complement medical treatment and do not replace it, and their effects cannot be expected without correcting obesity.

Additionally, a recent topic is the relationship between taste (umami) and hyperuricemia. Umami is one of the five basic tastes and is the sense that detects whether a food contains protein. Foods rich in purines often present an umami taste. Monosodium glutamate (MSG), one of the umami flavors, may directly induce obesity and metabolic syndrome through the formation of uric acid as well as fructose metabolism [53]. Thus, further research is needed to determine how purine-rich umami foods affect hyperuricemia [54].

## 6. Gout

Gout, distinguished by acute episodes of joint inflammation, occurs when there is an increase in serum uric acid levels of more than 7 mg/dL (420 µmol/L), contributing to the formation of deposits of monosodium urate (MSU), a tiny needle-shaped crystalline formation of uric acid [55]. Mendelian randomization studies suggest convincing evidence of an association with hyperuricemia exists for gout [56].

Gout can be a risk factor for CVD. A study showed that the risk for myocardial infarction and stroke in patients with gout was elevated 1.82 and 1.71 times, respectively, compared to those without gout [57]. Some studies have reported that CVD risk increases within 120 days, especially during the first 30 days following an acute gout flare [58,59]. Therefore, preventive medicine and gout management could be effective for both acute gout flare and CVD prevention.

Regarding the guidelines for gout management, they vary depending on the country. The 2020 American College of Rheumatology Guideline for the Management of Gout (ACR2020) recommends pharmacological treatment of hyperuricemia, uric acid-lowering treatment (ULT), for all patients with tophaceous gout or frequent gout flares with serum uric acid target levels of ≤5.0 mg/dL. It does not recommend treatment for patients without gout, even those with CVD risks [11]. In contrast, the Japanese Guideline on Management of Hyperuricemia and Gout (JGMHG) recommends lowering serum uric acid levels to ≤6 mg/dL and recommends pharmacological treatment for hyperuricemic patients (serum uric acid levels ≥ 8 mg/dL) with CKD and CVD risk and for hyperuricemic patients (serum uric acid levels ≥ 9 mg/dL) without CKD and CVD risk [12,60]. A recent retrospective cohort study using the JMDC Claims Database showed that occurrences of gout flare for both gout and asymptomatic hyperuricemia in patients who achieved serum uric acid levels ≤ 6.0 mg/dL with ULT decreased compared to patients whose serum uric acid levels remained >6.0 mg/dL or who were not receiving ULT [61]. Additionally, based on a meta-analysis of studies, a longer duration of ULT with achieving serum uric acid levels < 6 mg/dL was associated with reduced gout flares [62]. These studies provide evidence of serum uric acid levels ≤ 5.0 or 6.0 mg/dL as a treatment target for patients with gout and asymptomatic hyperuricemia.

From another perspective, MSU can be a better surrogate marker of gout flares. As a pathophysiology of gout flare, a previous prospective observational study found that an increase in MSU volume measured with dual-energy computed tomography was associated with a higher risk for gout flares [63]. In a prospective study examining the impact of ULT on MSU deposits in gout patients, the burden of MSU deposits significantly decreased over an average of 18 months of follow-up in patients undergoing lifestyle intervention and treated with allopurinol or febuxostat [64]. A change in MSU volume was significantly but weakly associated with a change in serum uric acid levels. No significant decline in MSU deposits was observed in patients who discontinued treatment. Recent research has shown that over a third of gout patients stop taking their ULT [65]. Allopurinol interrupters and discontinuers had indicators of more severe gout over time compared to adherers. These data indicate that the crucial aspects of gout management are continuing ULT and monitoring MSU deposits as a possible surrogate marker. Further research focusing on MSU deposits is required.

## 7. Diabetes Mellitus (Glucose Metabolism)

Serum uric acid levels are known to be influenced by the presence of diabetes mellitus and other lifestyle-related diseases. Epidemiological evidence to date indicates that type 2 diabetes mellitus is associated with gout [66]. On the other hand, it is also known that serum uric acid levels are not high in hyperglycemic conditions. According to past papers, both uric acid levels and the rate of hyperuricemia increase with increasing HbA1c levels but conversely decrease when HbA1c exceeds 6.0 to 6.9, indicating a bell-shaped relation due to the uricosuric effect of glucosuria [67]. The association between diabetes mellitus and serum uric acid levels is one of the most interesting topics.

Recently, Lee KW and Shin D reported that elevated serum uric acid levels may exacerbate the development of risk of type 2 diabetes mellitus in the Korean Genome and Epidemiology Study [68]. A total of 4152 Korean adults aged 45–76 years without type 2 diabetes mellitus, cancer, or gout at baseline in 2007–2008 were followed up until 2016. In this study, they reported that high levels of serum uric acid and high-sensitivity C-reactive protein (hsCRP) in combination were also associated with an increased incidence of type 2 diabetes mellitus compared to low levels of serum uric acid and hsCRP.

Jiahao Zhu et al. investigated bidirectional associations of type 2 diabetes mellitus and glycemic traits with plasma serum uric acid levels using a Mendelian randomization approach [69]. The associations of type 2 diabetes mellitus and fasting insulin with serum uric acid levels were shown. In addition, Xueting Hu et al. investigated the association between elevated plasma uric acid levels and a higher risk of insulin resistance in newly diagnosed type 2 diabetes through Mendelian randomization analysis. However, there was no strong association between uric acid and insulin resistance in this study [70].

On the other hand, the relationship between serum uric acid level and sodium–glucose cotransporter-2 (SGLT-2) inhibitors has also been attracting attention. Regardless of its precise mechanism, it has been known for a while that SGLT-2 inhibitors could significantly reduce serum uric acid levels in patients with type 2 diabetes mellitus^6^. Recently, Banerjee M et al. reported that SGLT2 inhibitors significantly reduced the risk of gout in individuals with type 2 diabetes mellitus and/or heart failure (HF) using data from five randomized controlled trials (RCTs) [71].

## 8. Metabolic Syndrome

Hyperuricemia is known to be associated with metabolic syndromes such as obesity, insulin resistance, and dyslipidemia. However, the causal relationship between uric acid and these metabolic syndromes remains unclear. The following studies have recently been reported and are expected to clarify the causal relationship between uric acid and metabolic syndrome in the future.

The relationship between hyperuricemia and dyslipidemia has long been recognized [72]. A recent longitudinal cohort study showed that triglyceride and HDL cholesterol levels and dyslipidemia were significantly associated with the development of hyperuricemia [73]. On the other hand, hyperuricemia is known to be associated with dyslipidemia [74], although a meta-analysis reported that treatment with allopurinol for 4 to 24 weeks did not significantly reduce serum triglyceride and low-density lipoprotein cholesterol levels [75]. Further studies, including long-term intervention trials, are needed to clarify the causal relationship between uric acid and dyslipidemia.

Although hyperuricemia is associated with the risk of NAFLD in a systematic review and meta-analysis [76], it remains unclear whether there is a bidirectional or temporal relationship between it and NAFLD. Using logistic regression and cross-lagged panel analysis, Zhimin Ma and colleagues showed a unidirectional relationship from hyperuricemia to NAFLD incidence [77]. This study suggests that hyperuricemia plays a fundamental role in the development of NAFLD. However, the effect of ULT on NAFLD has not been fully investigated [78]. Whether hyperuricemia is a therapeutic target for preventing the onset and progression of NAFLD needs to be investigated.

A meta-analysis of bariatric surgery showed that serum uric acid levels decreased from three months after surgery and persisted until the third year after surgery, as well as a reduction in the incidence of gout attacks. However, serum uric acid levels increased one month after surgery [79]. In addition, weight loss after bariatric surgery is associated with reduced serum uric acid levels. Bariatric surgery may be an important treatment option in preventing and managing hyperuricemia or gout.

An association between metabolism and XOR, a rate-limiting enzyme involved in the production of not only uric acid but also reactive oxygen species (ROS), has been reported [80]. Plasma XOR activity, determined via liquid chromatography/triple quadrupole mass spectrometry using radio-labeled xanthine, is associated with insulin resistance and glycemic control status [81]. Since plasma XOR activity has been reported to be associated with serum uric acid levels [82], the involvement of XOR activity should also be taken into account when considering uric acid and metabolic syndrome.

In experimental and clinical studies, the administration of benzbromarone, a non-selective inhibitor of URAT1, has been reported to increase adiponectin levels and improve insulin resistance [83,84]. Dotinurad, a URAT1 selective inhibitor, has been shown to ameliorate insulin resistance by attenuating hepatic steatosis and promoting brown adipose tissue re-browning in mice [85]. Further studies in humans are needed to clarify the effects of dotinurad on metabolism, including insulin resistance.

## 9. Atherosclerosis

Serum uric acid levels tend to be elevated by the presence of hypertension, CKD, and metabolic syndrome, all of which are established risk factors for atherosclerosis. Therefore, serum uric acid levels can be used as a useful biomarker for atherosclerosis. However, it remains a matter of debate whether hyperuricemia per se is an independent causal risk factor for atherosclerosis, such as endothelial dysfunction and arterial stiffening. Experimental studies have shown that hyperuricemia causes oxidative stress, inflammation, and dephosphorylation of endothelial nitric oxide synthase, which can lead to the progression of atherosclerosis and vascular dysfunction [86]. Indeed, observational clinical studies have shown that hyperuricemia is independently associated with the progression of atherosclerosis [87,88]. These findings indicate the possibility that hyperuricemia could be a causal risk factor for atherosclerosis. However, it has not been determined whether ULT inhibits the progression of atherosclerosis and deterioration of vascular function.

Allopurinol reduced the carotid intima-media thickness (CIMT) in hyperuricemic patients with type 2 diabetes [89] and recent ischemic stroke [90]. Meanwhile, in the recent PRIZE (program of vascular evaluation under uric acid control by xanthine oxidase inhibitor, febuxostat: multicenter, randomized controlled) study [91], 24 months of febuxostat, compared to non-pharmacological lifestyle modification care for hyperuricemia, did not delay the progression of CIMT in Japanese patients with asymptomatic hyperuricemia. Interestingly, in a sub-analysis of the PRIZE study, a greater reduction in serum uric acid was associated with an attenuation of CIMT progression, although no optimal target serum uric acid level to delay CIMT progression was observed [92].

Regarding the effects on vascular functional parameters, febuxostat did not change endothelial function as assessed by flow-mediated vasodilation [93], while the therapy modestly improved arterial stiffness markers involving pulse wave velocity (PWV) and the cardio-ankle vascular index (CAVI) [94]. A meta-analysis demonstrated that allopurinol did not affect arterial stiffness as assessed by PWV [95]. In addition, neither febuxostat nor topiroxostat had any obvious effects on arterial stiffness markers (PWV and CAVI) in patients with hypertension and treated hypertension [96]. Whether these conflicts about the effects of XOR inhibitors on atherosclerosis and vascular functional markers depend on the differences in the study design, population, or drug remains uncertain. Finally, since little clinical evidence on the effects of uricosuric agents, such as benzbromarone, on those markers is currently available, further research is required to assess this issue.

## 10. Hypertension

Epidemiological studies have suggested a significant association between elevated serum uric acid levels and hypertension. In several RCTs, uric acid-lowering medicine, including XOR inhibitors, showed a beneficial effect on blood pressure (BP), although some intervention studies reported no effect on BP. Therefore, some recent topics of uric acid research would be helpful to understand the relationship of serum uric acid with hypertension better, as follows.

First, numerous epidemiological studies have shown that higher serum uric acid levels predicted incident hypertension [97]. Serum XOR level was also associated with higher BP through generating ROS in cross-sectional studies [98]. Second, Feig DI et al. suggested that allopurinol reduced BP by both lowering systemic vascular resistance and plasma renin activity compared to placebo as the first intervention study of the effect of XOR inhibitor on BP [99]. A recent systematic review also suggested that allopurinol revealed a greater reduction of both systolic BP and diastolic BP [100]. Conversely, the FEATHER study (Febuxostat Versus Placebo Randomized Controlled Trial Regarding Reduced Renal Function in Patients with Hyperuricemia Complicated by Chronic Kidney Disease Stage 3) showed that febuxostat decreased BP, but there was no difference in BP reduction compared to placebo [101].

There are some points to be discussed in serum uric acid-hypertension association, which can explain the inconsistent study results of the above studies. First, a vast number of metabolic confounders are involved in uric acid research. For example, the serum uric acid-hypertension association varies depending on age, and it was stronger in children with few confounders [102]. Moreover, recent cross-lagged and mediation analyses revealed the role of BP on the association of serum uric acid with other diseases. Tian X et al. suggested that serum uric acid can elevate both systolic and diastolic BP, which partially facilitated the effect of serum uric acid on incident CVD (mediation effect: 57.6% for systolic BP and 46.3% for diastolic BP) [103]. Mendelian randomization mediation analysis in UK Biobank also supports the idea that higher BP mediates approximately one-third of the effect of serum uric acid on CVD risk [31]. These results may help us to elucidate the complicated network of serum uric acid and other confounders and estimate the effect of lowering BP on CVD prevention. Second, uric acid extracellularly acts as a strong antioxidant but intracellularly shows pro-inflammatory effects [104]. Intracellular uric acid, which is influenced by food very much, is more important in terms of an increased risk for vascular disease; however, it remains to be elucidated how uric acid-lowering medicines, including XOR inhibitors, affect extracellular and intracellular uric acid levels, respectively. Third, serum uric acid levels are also regulated by uric acid excretion. A retrospective study of 84 patients (mean age: 64 ± 16 years) suggested that systolic BP significantly decreased at 3 months after the start of dotinurad compared to baseline. The authors concluded that dotinurad could reduce systolic BP by possibly a relative inhibition of glucose transporter 9 (GLUT9) [105]. On the other hand, pegloticase, a recombinant uricase, significantly reduced mean arterial pressure for 6 months in patients with chronic gout, independent of changes in renal function [106]. The beneficial effect of uric acid-lowering medicines on BP may depend on how they work in the process of uric acid regulation.

Many epidemiological studies support the significant relationship between serum uric acid and hypertension. However, further RCTs are needed to clarify how its association varies depending on age, confounders or mediators, uric acid distribution, and regulation for an appropriate strategy to manage CVD risk.

### Hypertensive Disorders of Pregnancy

Hypertensive disorders of pregnancy (HDP) are defined as hypertension in pregnant women, which is classified according to the timing of hypertension diagnosis and the presence or absence of clinical findings such as proteinuria and organ damage as follows: chronic hypertension; white coat hypertension; masked hypertension; gestational hypertension; and pre-eclampsia (de novo and superimposed on chronic hypertension) [107]. It is very important to distinguish pre-eclampsia from other types of HDP in clinical practice despite all types of HDP being high-risk pregnancies [108].

Previous studies have indicated an association between uric acid and pre-eclampsia [109], and measurement of serum uric acid is recommended in patients with HDP by European and American guidelines [110,111]: Elevated uric acid levels occur in pregnancies with pre-eclampsia compared to normal pregnancies [112,113]. Several mechanisms of elevated uric acid levels in pre-eclampsia have been reported [109]. Firstly, vasoconstrictors such as angiotensin II reduce renal blood flow, leading to decreased uric acid excretion. Secondly, hypoxia associated with placental insufficiency increases ROS and oxidative stress, resulting in increased uric acid production and decreased uric acid excretion. In contrast, some studies have reported that elevated serum uric acid levels indicate the severity of the disease process in pre-eclampsia [109], and the ratio of serum uric acid to creatinine is associated with the development of pre-eclampsia and adverse pregnancy outcomes [114]. Furthermore, an observational cohort study revealed that the development of pre-eclampsia is associated with elevated serum uric acid levels before 20 weeks of gestation, especially during the early 8–12 weeks, and the effect diminishes with increasing gestational weeks, suggesting that elevated serum uric acid in early pregnancy may be a potential causative role in pre-eclampsia [115]. However, it is not yet clear whether uric acid is merely a risk marker or a cause that contributes to the progression of pre-eclampsia pathology, and future investigation is needed.

Women who have experienced HDP suffer higher rates of CVD events, including heart failure, coronary artery disease, and stroke [116]. Additionally, the offspring of women with hypertensive pregnancy are more likely to suffer from CVD [117]. Uric acid may play an important role in the link between HDP and the subsequent development of CVD in both mother and child.

## 11. Kidney Diseases

The kidney plays a crucial role in regulating serum uric acid levels, accounting for 60–70% of uric acid excretion. Normally, the kidney reabsorbs approximately 90% of the uric acid filtered by the glomerulus in the proximal tubules [118]. Gout is present in one-third of patients with CKD, and a “gouty nephropathy” resulting from MSU deposits could be a significant cause of CKD [119]. A study reported that patients with gout are 29 percent more likely to suffer from advanced CKD and more than 200 percent more likely to have kidney failure [120]. Additionally, another study noted that patients with severe gout exhibited a diffuse hyperechoic kidney medulla pattern [121]. Therefore, gout serves as a warning sign for CKD or indicates a higher risk for kidney disease. However, the precise role of uric acid in CKD has not been completely determined. Hyperuricemia is associated with hypertension and aging and with renal atherosclerosis in patients with CKD, according to a cross-sectional analysis [122]. Although many epidemiologic studies have reported that elevated serum uric acid is a predictor of the development of CKD, the causal relationship remains controversial. The relationship between hyperuricemia and kidney disease was described by the International Kidney Disease (KDIGO) in its 2012 “CKD Guidelines” [123]. Although the guidelines indicate the importance of managing hyperuricemia in CKD, the evidence for the use of uric acid-lowering agents for renal protection in CKD without gout is insufficient and not recommended. In 2015, an RCT of patients with CKD stage G3–4 and asymptomatic hyperuricemia showed that 6 months of febuxostat treatment reduced renal dysfunction compared to placebo [124]. In 2018, the results of two RCTs of febuxostat and topiroxostat for hyperuricemia were reported [101,125]. In the FEATHER study of CKD stage G3 patients with asymptomatic hyperuricemia, febuxostat treatment in the setting of adequate CKD treatment showed no significant effect in preventing renal dysfunction [101]. However, a post-hoc analysis of the FEATHER study reported a significantly higher mean estimated glomerular filtration rate (eGFR) slope in the febuxostat group than in the placebo group in CKD patients without proteinuria [126]. In the UPWARD study of diabetic nephropathy patients with hyperuricemia (with/without gout), ULT with topiroxostat significantly reduced eGFR decline compared to placebo, although there was no significant difference in albuminuria [125]. In 2019, the FREED (Febuxostat for Cerebral and CaRdiorenovascular Events PrEvEntion StuDy) showed a significant reduction in renal dysfunction, including proteinuria, in the febuxostat treatment group [127]. The PERL (Preventing Early Renal Loss in Diabetes) study and CKD-FIX (Controlled Trial of Slowing of Kidney Disease Progression from the Inhibition of Xanthine Oxidase) showed that allopurinol-assisted ULT did not slow the decline in eGFR compared with placebo [128,129]. However, it is important to note that both the PERL and CKD-FIX included a large number of participants with normal uric acid levels. In this regard, it is important to note that lowering uric acid levels in normouricemic patients should not be included in clinical trials investigating the effects of hyperuricemia in CKD, as normouricemia is not associated with CKD progression. A meta-analysis including these RCTs was reported in 2020 and found that ULT did not reduce renal failure events (such as a 30% decline in eGFR during follow-up, doubling of serum creatinine levels, and renal failure) but reduced GFR decline (weighted mean difference, 1.18 mL/min/1.73 m^2^/year; 95% confidence interval, 0.44–1.91) [130]. Taken together, these results suggest that ULT is probably not indicated for all patients with CKD and that future clinical trials should be conducted in specific subgroups, such as younger patients and those with nephrosclerosis and hyperuricemia.

## 12. Cardiovascular Diseases (CVD)

Uric acid is an end-product of purine metabolism in humans that is mainly regulated through the XOR pathway [131]. The activation of the XOR pathway generates oxidative stress and uric acid, causing vascular inflammation, which may play a role in developing CVD (Figure 3) [132]. Ample evidence has suggested that there is a plausible link between hyperuricemia and/or gout and worsening prognosis in patients with overt CVD [132]. This chapter will summarize the recent studies that have highlighted the clinical significance of hyperuricemia/gout, particularly in the fields of HF, ischemic heart disease (IHD), and arrhythmia.

### 12.1. Heart Failure

HF patients are more likely to associate hyperuricemia [133]. Additionally, there has been evidence that hyperuricemia is associated with an increased risk of incident HF and adverse outcomes [134]. However, the guidelines currently do not recommend ULT to improve the prognosis in patients with hyperuricemia and HF. As shown before, although it is possible that XOR inhibition may have clinical benefits in patients with symptomatic HF [132], previous prospective interventional studies with XOR inhibitors have not yet reported an improvement in HF outcomes. For instance, one of the XOR inhibitors, oxypurinol, did not improve clinical outcomes in unselected patients with moderate-to-severe HF in the OPT-CHF (The Efficacy and Safety Study of Oxypurinol Added to Standard Therapy in the Patients With New York Heart Association Class III-IV Congestive Heart Failure) study [135]. Contrary to the result, post-hoc analysis revealed that high serum uric acid (≥9.5 mg/dL) received improvement in clinical status by taking oxypurinol compared with placebo [135]. Another XOR inhibitor, allopurinol, failed to improve clinical status in patients with HF with reduced ejection fraction (≤40%) and elevated serum uric acid levels (≥9.5 mg/dL) in the EXACT-HF (the Xanthine Oxidase Inhibition for Hyperuricemic Heart Failure Patients) trial [136]. A systematic review and meta-analysis regarding the effect of ULT on patients with HF did not reveal any improvement in ejection fraction, B-type natriuretic peptide, 6 min walk test, all-cause mortality, and CVD death with ULT compared with placebo [137].

Thus, the efficacy of ULT in HF patients has not been previously determined, although the results of the FAST (Febuxostat versus Allopurinol Streamlined Trial) showed the possibility that febuxostat could have favorable effects for HF compared to allopurinol [138]. The LEAF-CHF (Effect of Urate-LowEring Agent Febuxostat in Chronic Heart Failure Patients with Hyperuricemia) study, which evaluates the improvement of plasma B-type natriuretic peptide using febuxostat in chronic HF patients with reduced ejection fraction and hyperuricemia, is currently in progress [139]. Clinical evidence needs to be established.

### 12.2. Ischemic Heart Disease

Albeit with the excellent predictive value of hyperuricemia for patients with IHD [140], whether or not ULT allows CVD risk reduction for this population remains an open question. It has been reported that high-dose allopurinol (600 mg/day) exerts beneficial effects on exercise tolerance in patients with stable chronic angina pectoris [141]. Indeed, allopurinol significantly increased total exercise time and the time to chest pain from a baseline when compared to placebo [141]. Most recently, the ALL-HEART study was conducted to determine whether ULT with allopurinol improves major CVD outcomes in patients with IHD [142]. The ALL-HEART study enrolled 5721 patients aged 60 years or older with IHD, irrespective of serum uric acid levels. While allopurinol (600 mg/day) profoundly decreased serum uric acid levels (from 0.34 mmol/L to 0.18 mmol/L), there was no difference in major CVD outcomes between the allopurinol group and the usual care group [142]. One of the major criticisms was that the ALL-HEART study did not include patients with a history of gout. In this regard, a retrospective observational study has underlined that preceding gout flare was more prevalent in patients who experienced a CVD event as compared with those who did not experience it [58].

MSU is frequently observed in synovial fluid of gout patients. MSU exacerbates gout-associated inflammation through inflammasome activation and interleukin-1β secretion (Figure 2) [143]. A recent study reported that dual-energy computed tomography makes it possible to identify MSU depositions in human aorta and coronary arteries in vivo [144]. Future studies with a novel imaging approach can be encompassed to explore the role of MSU in living patients.

### 12.3. Arrhythmia

In 2010, the first association between hyperuricemia and atrial fibrillation (AF), one of the most common and clinically important arrhythmias, was reported [145]. Subsequent studies have consistently validated this association, particularly in the incidence of new-onset AF [146,147,148]. A recent comprehensive meta-analysis involving 608,810 participants from 11 studies confirmed the increased risk of incident AF in individuals with hyperuricemia (risk ratio, 2.42; 95% confidence interval (CI), 1.24–3.03) across countries [149]. Moreover, serum uric acid concentrations also serve as a valuable marker for AF recurrence following AF catheter ablation [150]. Another meta-analysis involving 2046 patients from 14 studies demonstrated that individuals who experienced AF recurrence had a higher serum uric acid level compared to those who did not experience it (weighted mean difference, 0.69 mg/dL; 95% CI, 0.46–0.91) [151]. Furthermore, elevated serum uric acid was significantly associated with a higher AF recurrence rate after AF catheter ablation (odds ratio, 2.21; 95% CI, 1.73–2.83) [151]. These collective findings highlight the potential of elevated serum uric acid as a useful tool for stratifying risk in both new-onset AF and AF recurrence post-catheter ablation.

While investigations remain limited, some studies have reported associations between serum uric acid and other types of arrhythmic disorders, including ventricular arrhythmias [152] and atrioventricular block [153]. Such associations necessitate more robust and extensive validation through further research. A recent Mendelian randomization study supports the causal relationship between elevated serum uric acid and increased risk of CVD death, especially sudden cardiac death [30]. This finding is congruent with several prior studies examining patients with different backgrounds [152].

Despite these connections, it is critical to acknowledge the existence of conflicting evidence surrounding the potential benefits of ULT in the reduction of arrhythmias [154,155]. Further research is imperative to ascertain the effectiveness and usefulness of ULT definitively. In summary, while serum uric acid has been recognized as a significant indicator for AF and other arrhythmias, the therapeutic potential of ULT warrants further and detailed exploration.

## 13. Neurological Diseases

The effects of uric acid in neurological diseases are likely to vary greatly between stroke, a vascular disease, and Parkinson’s and Alzheimer’s diseases, which are neurodegenerative diseases [156]. A systematic review of umbrella reviews revealed a significant association of high serum uric acid levels with a decreased risk of several neurological diseases (i.e., Parkinson’s disease, Alzheimer’s disease, dementia, multiple sclerosis, neuromyelitis optica, and amyotrophic lateral sclerosis) [157]. Several recent studies that focused on the relationship between uric acid and varying neurological diseases are summarized below.

### 13.1. Stroke

There seems to be no doubt about the relationship between serum uric acid and stroke [158,159]. The Reasons for Geographic and Racial Differences in Stroke (REGARDS) study, a case-cohort study with large data sets, concluded that hyperuricemia may be a risk factor for stroke [160]. A meta-umbrella review showed class I evidence that high uric acid levels were associated with risk for stroke mortality [157]. However, some epidemiological studies reported inconsistent findings on the relationship between serum uric acid levels and stroke [161,162]. In addition, Jiaqi et al. conducted a prospective cohort study in Japan and showed that elevated serum uric acid levels are an independent predictor of total and ischemic stroke in women only [163]. Therefore, the causality remains controversial.

A retrospective cohort study conducted using the Taiwanese population-based National Health Insurance Research Database showed that gout patients treated with ULT had a lower risk of hospitalized stroke and all-cause mortality compared to those without ULT. In addition, the effect of uricosuric agents on reducing the risk of hospitalization due to stroke showed a dose-response relationship [164]. The ALL-HEART did not differ in the outcome of nonfatal stroke between participants assigned to allopurinol therapy and those assigned to usual care [142]. Thus, there is still little clear evidence of stroke prevention by ULT.

### 13.2. Parkinson’s Disease

Parkinson’s disease is the second most common neurodegenerative disease after Alzheimer’s disease. Serum uric acid levels are known to be low in Parkinson’s disease. Furthermore, lower serum uric acid levels have been associated with the risk of developing Parkinson’s disease, severity, nonmotor symptoms, and slow disease progression [165,166]. This mechanism is thought to be due to low uric acid levels, which inhibit oxidative stress that predisposes to dopaminergic neuron degeneration in Parkinson’s disease [167]. To demonstrate this mechanism, a clinical trial was conducted in early Parkinson’s disease patients receiving inosine to increase serum uric acid levels. However, no difference in the rate of clinical disease progression in Parkinson’s disease was observed compared to placebo [168].

### 13.3. Alzheimer’s Disease and Dementia

The relationship between uric acid and dementia, especially neurodegenerative diseases such as Alzheimer’s, has been studied. Higher serum uric acid levels were associated with better cognitive function and appeared to be neuroprotective. A prospective cohort study conducted in Sweden involving women only over a 44-year period showed an association between higher serum uric acid levels and a lower risk of dementia. This study supports the hypothesis that serum uric acid levels play a protective role in the development of dementia, regardless of dementia subtype [169]. A cross-sectional study obtained from the ReGAl 2.0 project in Italy also showed that serum uric acid levels were lower in patients with Alzheimer’s disease [170]. In contrast, the Atherosclerosis Risk in Communities (ARIC) cohort showed no association between serum uric acid levels and incident dementia [171]. Since evidence from observational studies is susceptible to numerous biases, the effects of gender, age, and dementia subtypes must also be considered. To date, few clinical interventional studies have investigated the potential of inosine therapy in patients with Alzheimer’s disease. However, basic evidence suggests that inosine might be a promising therapeutic strategy for Alzheimer’s disease thanks to its ability to modulate different brain mechanisms involved in neuroprotection [172].

### 13.4. Multiple Sclerosis and Neuromyelitis Optica

Multiple sclerosis and neuromyelitis optica are autoimmune central nervous system diseases; a meta-analysis of 10 case-control studies found that patients with multiple sclerosis and neuromyelitis optica had lower serum uric acid levels compared to healthy controls [173]. In a 12-month randomized, placebo-controlled trial following patients after inosine administration, it was determined that inosine did not possess neuroprotective effects, thus proving ineffective for relapsing–remitting multiple sclerosis. Nevertheless, a comprehensive evaluation of the efficacy of inosine for multiple sclerosis is needed, involving an extended follow-up period of over one year [174].

### 13.5. Amyotrophic Lateral Sclerosis

A national database of South Korea study found that the prevalence of amyotrophic lateral sclerosis (ALS) in gout patients was substantially lower than in the general population [175]. A cross-sectional study was conducted to determine the relationship between serum uric acid levels and cognitive impairment in patients with ALS in China. The results showed that a low serum uric acid level was an independent risk factor for cognitive impairment in patients with amyotrophic lateral sclerosis [176]. Furthermore, a longitudinal cohort study in China found an inverse association between serum uric acid levels and risk of death, particularly pronounced in male patients with ALS [177].

## 14. Malignancies

The relationship between malignancy and hyperuricemia requires multifaceted references to antioxidant effects and etiology. Although it has long been proposed that uric acid, particularly as an endogenous antioxidant, may exert anticarcinogenic properties [7,178], recently, some studies have been conducted on cancer risk of hyperuricemia and inflammation caused by gout, or oxidative stress involved in ROS, in the etiology of malignant tumors [179].

The association between uric acid levels and prognosis has also been examined in several papers on hematologic malignancies and solid tumors. It has been suggested that high uric acid levels are a poor prognostic factor in acute myeloid leukemia [180]. In diffuse large B-cell lymphoma, a high uric acid level is a shorter progression-free and overall survival [181].

Tumor lysis syndrome (TLS) is a life-threatening complication that is caused by the sudden and massive death of cancer cells upon chemotherapy [182,183,184] (Figure 4). According to the Japanese Adverse Drug Event Report database, the incidence of TLS has increased 7.5 times from 2003 to 2019, with a total of 620 cases reported. The mortality rate due to TLS is reported to be 15–30%, making it one of the oncologic emergencies for which preventive treatment is recommended in the guidelines. Intracellular metabolites, which include proteins, potassium, phosphorus, and nucleic acids, are rapidly released from lysed cells to circulating blood. This may result in hyperkalemia, hyperphosphatemia, and hyperuricemia, thereby inducing renal insufficiency, cardiac arrhythmias, seizures, neurological disorders, and ultimately death. As far as TLS treatment, the use of rasburicase and febuxostat has led to a breakthrough in ULT, guidelines have been developed, and a consensus has been reached.

In the past, TLS was mainly concerned with acute lymphocytic leukemia, aggressive lymphomas in advanced stages, and acute myeloid leukemia with high tumor volume. In recent years, however, caution has been exercised in cases of TLS in chronic lymphocytic leukemia and acute myeloid leukemia in the elderly with the advent of B-cell lymphoma-2 (BCL2) inhibitors, and in multiple myeloma, where multidrug combination therapy has become the mainstay with the advent of proteasome inhibition, immunomodulatory agents, and antibody drugs. While the risk of TLS in these cases has not been systematically examined, Howard et al. identified published Phase I–III clinical trials of monoclonal antibodies [185]. According to the paper, the incidence of TLS with alvocidib was 4.2% in recurrent treatment-resistant ALL, 13.2% in mantle cell lymphoma, indolent B-cell lymphoma, and chronic lymphocytic leukemia (CLL), and 42.2% in acute myeloid leukemia.

The relationship between uric acid and malignancies is uncertain. Therefore, it is necessary to continue to investigate the mechanism of how hyperuricemia is associated with the development of malignancy and to continue to manage TLS that may develop following chemotherapy. We must remember to evaluate and control TLS as a side effect prior to the use of new anticancer drugs.

Since there is no uniformity in the selection and use of uric acid-lowering medicine for TLS prophylaxis in clinical studies, adequate TLS management should also be evaluated when assessing the frequency of each TLS incidence.

## 15. Conclusions

Taking into account recent research on uric acid, it is important to acknowledge that while numerous studies have identified associations between uric acid and various diseases, establishing a definitive causal relationship remains challenging in numerous instances. Given the intricate and multifaceted nature of uric acid’s involvement across many diseases, there is a clear need for conducting additional comprehensive investigations into uric acid. One proposed solution to address these challenges is the implementation of high-quality prospective RCTs while considering their intricate interactions and potential implications. Such research endeavors are essential, as they hold the key to unraveling the complexities surrounding uric acid and revealing its potential role in disease pathogenesis. 

## Figures and Tables

**Figure 1 biomolecules-13-01519-f001:**
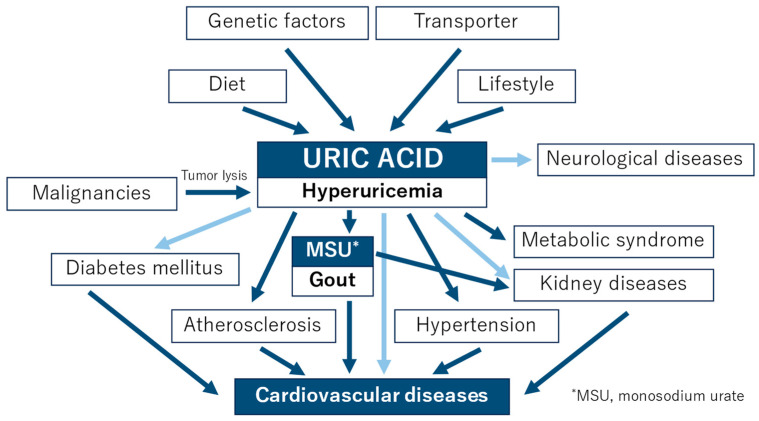
An overview of this review.

**Figure 2 biomolecules-13-01519-f002:**
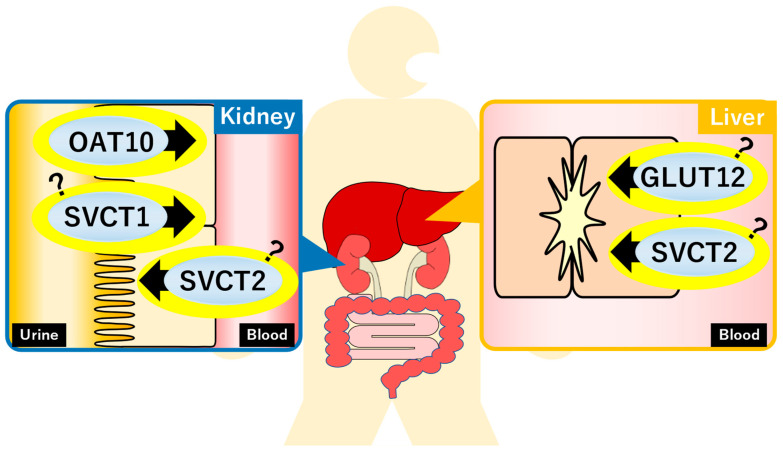
Recently identified uric acid transporters and their estimated functions. GLUT, glucose transporter; OAT, organic anion transporter; SVCT, sodium-dependent vitamin C transporter.

**Figure 3 biomolecules-13-01519-f003:**
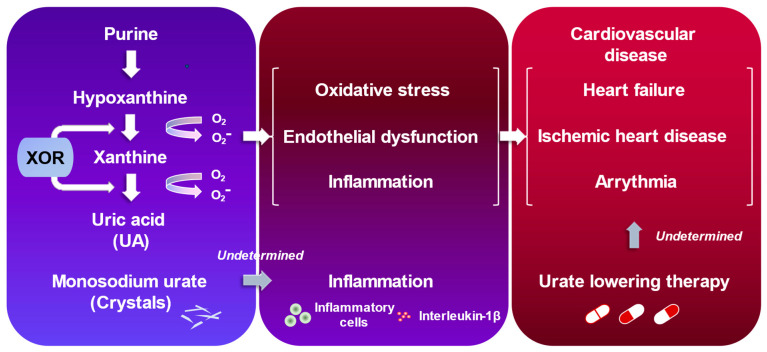
Uric acid and cardiovascular diseases.

**Figure 4 biomolecules-13-01519-f004:**
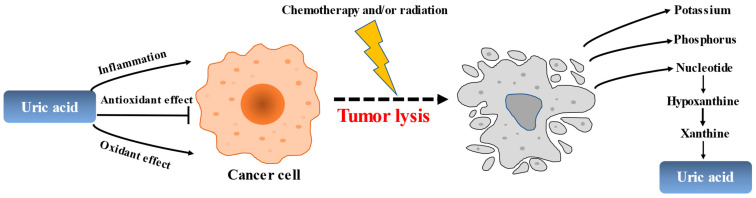
The role of uric acid in the pathogenesis of tumors and tumor lysis.

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
