# Peer review of "Exploring the Multifaceted Nexus of Uric Acid and Health: A Review of Recent Studies on Diverse Diseases"

_biomolecules, 2023, doi:10.3390/biom13101519_

Round 1
Reviewer 1 Report
This excellent review summarized ten domains associated with hyperuricemia and gout: diet, lifestyle, diabetes mellitus, metabolic diseases, atherosclerosis, hypertension, cardiovascular disease, neurologic diseases, malignancies, and transporters. It gives readers a complete picture of how hyperuricemia and gout affect health.
The authors mentioned that genetic factors are recognized as primary contributors to hyperuricemia and gout. However, they did not provide a detailed description. It is better to include them in the review.
Some blue lines in Figure 1 crossed the boxes. The thin lines can be used to avoid overlap with the boxes. A line is missing between transporter and gout.
As blood is filtered through the kidney, uric acid can build up and form urate crystals in the kidney, which may lead to kidney disease and failure over time, especially if the gout is left untreated. The relationship between gout and kidney disease should be included in this review.
Some experiments have reported that uric acid is an antioxidant that may protect against human aging and oxidative stress. Can the authors discuss them in this review as well?
Author Response
This excellent review summarized ten domains associated with hyperuricemia and gout: diet, lifestyle, diabetes mellitus, metabolic diseases, atherosclerosis, hypertension, cardiovascular disease, neurologic diseases, malignancies, and transporters. It gives readers a complete picture of how hyperuricemia and gout affect health.
Response: Thank you for your kind summary of our manuscript.
The authors mentioned that genetic factors are recognized as primary contributors to hyperuricemia and gout. However, they did not provide a detailed description. It is better to include them in the review.
Response: Thank you for your helpful suggestion. We have incorporated a new section on genetic factors, providing a detailed explanation. Additionally, we have removed the term ‘primary’ as it is challenging to definitively determine whether genetic factors or lifestyle factors hold greater importance. We recognize that the relative significance of genetic factors versus lifestyle factors can vary among individuals.
Some blue lines in Figure 1 crossed the boxes. The thin lines can be used to avoid overlap with the boxes. A line is missing between transporter and gout.
Response: Thank you for your valuable suggestion. We have made revisions to the Figure based on your recommendations. We have added a new section on gout and included 'gout' in the Figure.
As blood is filtered through the kidney, uric acid can build up and form urate crystals in the kidney, which may lead to kidney disease and failure over time, especially if the gout is left untreated. The relationship between gout and kidney disease should be included in this review.
Response: We appreciate your suggestion, and we have taken it into consideration. New sections on gout and kidney diseases have been added to this review, and we have also included these aspects in the Figure
Some experiments have reported that uric acid is an antioxidant that may protect against human aging and oxidative stress. Can the authors discuss them in this review as well?
Response: Thank you for your kind suggestion. We added some explanations of antioxidant effects of uric acid in introduction sections below.
The activity of uricase, an enzyme catalyzing conversion of uric acid to allantoin, was lost about 8-20 million years ago, and therefore, uric acid is the end product of purine me-tabolism in humans.[6] In addition, most uric acid filtered in the glomeruli is reabsorbed by the renal tubules, resulting in 5 to 10 times higher concentrations of serum uric acid in humans than those in other mammalians. These findings suggest that uric acid was nec-essary for human evolution. Experimental studies have shown that uric acid is a powerful antioxidant.[7] Uric acid may exert its beneficial effects through protecting cells from oxi-dative damage by maintaining superoxide dismutase, scavenging radical species, and chelating transition metals.[8,9] However, the worldwide prevalence of patients with hyperuricemia or gout is increasing.
Reviewer 2 Report
Comments on the revised "Exploring the Multifaceted Nexus of Uric Acid and Health: A Review of Recent Studies". Manuscript ID biomolecules-2614725-v1
The authors investigated the relationship between uric acid and ten specific domains: diet, lifestyle, diabetes mellitus, metabolic diseases, atherosclerosis, hypertension, cardiovascular disease, neurological diseases, malignancies, and transporters from recent studies
There are several comments:
0. A review should be honesty to the data. The authors should not propagate the good of ULT.
1. An important domain, CKD, is not mentioned. Is it because the evidence of ULT in CKD is negative ?
2. Some domains overlap with each other.
3. Metabolic disease is a loose domain. Focusing on hyperlipidemia may be clearer. Or “metabolic syndrome” is a better title.
4. New studies should be put into the context of literature.
5. Lack of meta-analysis data is an important weakness of this review.
Several meta-anlysis had been published about the effect of ULT on BP, MACE, renal function, heart failure and mortality.
It is weak to say that “Based on the results of the FAST, it is possible that febuxostat will probe to be effective in HF patients”. There were already several small trials. The authors should report.
6. Tumor lysis is a specific phenomenon and not related to usual practice.
7. New transporters should be put into a renal tubule transporter model. Ordinary readers could not get anything from this part. If the authors did not talk about URAT1, GLUT9, OATs, SGLT2 and the drugs on these transporters, it seems too far to talk about these new transporters without definite influence on clinical practice.
8. Low UA is sometimes with risk. Low serum uric acid levels were associated with increased risk of Parkinson’s Disease.
9. The authors should be balanced in reports. For example, meta-analysis do not demonstrate benefit of ULT for CKD and HF. It seems clear that after the hype of early ULT trials, large RCTs do not support the benefits of ULT for CKD.
Some literatures of meta-analysis:
Effect of Urate-Lowering Therapy on Cardiovascular and Kidney Outcomes
Non-genetic risk and protective factors and biomarkers for neurological disorders: a meta-umbrella systematic review of umbrella reviews
Effect of Uric Acid-Lowering Agents on Patients With Heart Failure: A Systematic Review and Meta-Analysis of Randomised Controlled Trials
In conclusion, the authors did a large review of uric acid in several domains. However, larger is not better. The authors should have a focus and have to summarize scientific evidence unbiasedly.
Author Response
The authors investigated the relationship between uric acid and ten specific domains:
diet, lifestyle, diabetes mellitus, metabolic diseases, atherosclerosis, hypertension cardiovascular disease, neurological diseases, malignancies, and transporters from recent studies
Response: Thank you for your kind summary of our manuscript.
There are several comments:
- A review should be honesty to the data. The authors should not propagate the good of ULT.
Response: Thank you for your comments. We have carefully revised our article to take a more neutral position, incorporating recent data including systematic reviews and meta-analyses as your recommendations. We did not propagate the good of ULT, except for preventing gout and hypertension.
- An important domain, CKD, is not mentioned. Is it because the evidence of ULT in CKD is negative?
Response: Thank you for your important comments. We have incorporated a new section on CKD, providing a detailed explanation including recent evidence of ULT in CKD.
- Some domains overlap with each other.
Response: Thank you for your valuable comments. We have carefully checked our manuscript and revised some overlap domains, such as dotinurad, xanthine oxidoreductase (XOR), and so on.
- Metabolic disease is a loose domain. Focusing on hyperlipidemia may be clearer. Or “metabolic syndrome” is a better title.
Response: Thank you for your suggestions. We have changed the term of “metabolic disease” to “metabolic syndrome”
- New studies should be put into the context of literature.
Response: Thank you for your kind suggestions. We have added new studies with context such as the FEATHER study, UPWARD study, FREED, PERL, CKD-FIX, and others. However, due to page limitations, we could not provide detailed information on all the studies.
- Lack of meta-analysis data is an important weakness of this review.
Several meta-anlysis had been published about the effect of ULT on BP, MACE, renal function, heart failure and mortality.
It is weak to say that “Based on the results of the FAST, it is possible that febuxostat will probe to be effective in HF patients”. There were already several small trials. The authors should report.
Response: Thank you for your valuable suggestions. We have incorporated several meta-analysis articles, and we have revised the sentence about febuxostat below as per your recommendations, which strengthens our article. We greatly appreciate your kind suggestions.
“the efficacy of ULT in HF patients has not been previously determined, although the results of the FAST (Febuxostat versus Allopurinol Streamlined Trial) showed the possibility that febuxostat could have favorable effects for HF compared to allopurinol.”
- Tumor lysis is a specific phenomenon and not related to usual practice.
Response: Thank you for your comments. We agree with your suggestions and have revised the first half of the section to show the relationship between uric acid and malignancies, as well as to explain that TLS is a uric acid-related disease with the potential for serious complications.
- New transporters should be put into a renal tubule transporter model. Ordinary readers could not get anything from this part. If the authors did not talk about URAT1, GLUT9, OATs, SGLT2 and the drugs on these transporters, it seems too far to talk about these new transporters without definite influence on clinical practice.
Response: Thank you for your valuable comments. We have incorporated explanations of well-known transporters, as per your recommendations. Our articles focused on recent update in uric acid research and introduces novel transporters.
- Low UA is sometimes with risk. Low serum uric acid levels were associated with increased risk of Parkinson’s Disease.
Response: Thank you for your important comments. We have explained the positive effects of high serum uric acid for neurological diseases. We added some explanations.
- The authors should be balanced in reports. For example, meta-analysis do not demonstrate benefit of ULT for CKD and HF. It seems clear that after the hype of early ULT trials, large RCTs do not support the benefits of ULT for CKD.
Some literatures of meta-analysis:
Effect of Urate-Lowering Therapy on Cardiovascular and Kidney Outcomes
Non-genetic risk and protective factors and biomarkers for neurological disorders: a meta-umbrella systematic review of umbrella reviews
Effect of Uric Acid-Lowering Agents on Patients With Heart Failure: A Systematic Review and Meta-Analysis of Randomised Controlled Trials
Response: Thank you for your kind recommendations. We have added a new section on kidney diseases and incorporated references to relevant meta-analyses, as per your kind suggestions. We believe that these revisions have made the article more balanced and robust.
In conclusion, the authors did a large review of uric acid in several domains. However, larger is not better. The authors should have a focus and have to summarize scientific evidences unbiasedly.
Response: Thank you for your important comments. This review article aims to expand the interest of readers in the field of uric acid research. By comprehensively addressing the associations between uric acid and various diseases, we summarized the existing knowledge and highlights areas that warrant future investigation. As a result, it encompasses a broad spectrum of topics, and we hope this aspect is understood and appreciated.
Reviewer 3 Report
The review by Kuwabara and colleagues summarize current literature on the UA physiology and pathology. The paper is overall well written and of great interest and Authors are well recognized experts in the topic. My comments are mostly minor:
- I would change the title to include the focus Authors put on disease. Maybe "...A review of recent studies on diverse diseases".
- I would move the section on urate transporters to the introduction section
- I believe Authors should include one important disease in which uric acid plays a central role and for which its measurement is recommended by international guidelines: preeclampsia and in general hypertensive disorders of pregnancy. This is not only relevant during pregnancy, since women who suffered from hypertensive disorders of pregnancy are at increased risk of CVD during their entire life. Furthermore their offspring are also at increased CV risk. Uric acid may represent the trade-union between hypertensive disorders of pregnancy and later development of CVD, thus it is of crucial importance. Finally, recently uric acid to creatinine ratio, which normalizes uric acid for kidney function and somehow help stratify patients who are more likely to be UA hyperproducers , has shown to be associated with preeclampsia development.
Author Response
The review by Kuwabara and colleagues summarize current literature on the UA physiology and pathology. The paper is overall well written and of great interest and Authors are well recognized experts in the topic. My comments are mostly minor:
Response: Thank you for your kind summary of our manuscript.
- I would change the title to include the focus Authors put on disease. Maybe "...A review of recent studies on diverse diseases".
Response: Thank you for your kind suggestion. We have revised the title to “Exploring the Multifaceted Nexus of Uric Acid and Health: A Review of Recent Studies on Diverse Diseases".
- I would move the section on urate transporters to the introduction section
Response: Thank you for your valuable suggestion. We have moved the section on urate transporters to the first section as your kind recommendation.
- I believe Authors should include one important disease in which uric acid plays a central role and for which its measurement is recommended by international guidelines: preeclampsia and in general hypertensive disorders of pregnancy. This is not only relevant during pregnancy, since women who suffered from hypertensive disorders of pregnancy are at increased risk of CVD during their entire life. Futhermore their offspring are also at increased CV risk. Uric acid may represent the trade-union between hypertensive disorders of pregnancy and later development of CVD, thus it is of crucial importance. Finally, recently uric acid to creatinine ratio, which normalizes uric acid for kidney function and somehow help stratify patients who are more likely to be UA hyperproducers, has shown to be associated with preeclampsia development.
Response: Thank you very much for your valuable suggestions. We have added an explanation on “Hypertensive disorders of pregnancy” in Hypertension section below.
Hypertensive disorders of pregnancy
Hypertensive disorders of pregnancy (HDP) are defined as hypertension in pregnant women, which is classified according to the timing of hypertension diagnosis and the presence or absence of clinical findings such as proteinuria and organ damage as follows: chronic hypertension; white coat hypertension; masked hypertension; gestational hyper-tension; pre-eclampsia (de novo and superimposed on chronic hypertension) [107]. It is very important to distinguish pre-eclampsia from other types of HDP in clinical practice, despite all types of HDP are high-risk pregnancies [108].
Previous studies have indicated an association between uric acid and pre-eclampsia [109], and measurement of serum uric acid is recommended in patients with HDP by Eu-ropean and American guidelines [110,111]: Elevated uric acid levels occur in pregnancies with pre-eclampsia compared to normal pregnancies [112,113]. Several mechanisms of el-evated uric acid levels in pre-eclampsia have been reported [109]. Firstly, vasoconstrictors such as angiotensin Ⅱ reduce renal blood flow, leading to decreased uric acid excretion. Secondly, hypoxia associated with placental insufficiency increases ROS and oxidative stress, resulting in increased uric acid production and decreased uric acid excretion. In contrast, some studies have reported that elevated serum uric acid levels indicate the se-verity of the disease process in pre-eclampsia [109], and the ratio of serum uric acid to cre-atinine is associated with the development of pre-eclampsia and adverse pregnancy out-comes [114]. Furthermore, an observational cohort study revealed that the development of pre-eclampsia is associated with elevated serum uric acid levels before 20 weeks of gesta-tion, especially during the early 8-12 weeks, and the effect diminishes with increasing ges-tational weeks, suggesting that elevated serum uric acid in early pregnancy may be a po-tential causative role in pre-eclampsia [115]. However, it is not yet clear whether uric acid is merely a risk marker or a cause that contributes to the progression of the pre-eclampsia pathology, and future investigation is needed.
Women who have experienced HDP suffer higher rates of CVD events including heart failure, coronary artery disease, and stroke [116]. Additionally, the offspring of women with hypertensive pregnancy are more likely to suffer from CVD [117]. Uric acid may play an important role in the link between HDP and the subsequent development of CVD in both mother and child.
Round 2
Reviewer 1 Report
The revised manuscript is excellent and I do not have additional comments.
Reviewer 2 Report
The authors had a significant improvement in all our comments.
This review becomes more extensive than the previous one.
The scientific evidence is unbiasedly provided.
The new meta-analysis data are mentioned.
Reviewer 3 Report
Authors appropriately revised the manuscript according to my suggestions. I commend the Authors for their nice work.